# A Pilot Investigation of Visual Pathways in Patients with Mild Traumatic Brain Injury

Paul Harris [1] and Mark H. Myers [2],*

1   Department of Vision Therapy, Southern College of Optometry, Memphis, TN 38104, USA
2   Department of Anatomy and Neurobiology, University of Tennessee Health Sciences Center,
    Memphis, TN 38163, USA
*   Correspondence: mhmyers99@gmail.com

**Abstract:** In this study, we examined visual processing within primary visual areas (V1) in normal and visually impaired individuals who exhibit significant visual symptomology due to sports-related mild traumatic brain injury (mTBI). Five spatial frequency stimuli were applied to the right, left and both eyes in order to assess the visual processing of patients with sports-related mild traumatic brain injuries who exhibited visual abnormalities, i.e., photophobia, blurriness, etc., and controls. The measurement of the left/right eye and binocular integration was accomplished via the quantification of the spectral power and visual event-related potentials. The principal results have shown that the power spectral density (PSD) measurements display a distinct loss in the alpha band-width range, which corresponded to more instances of medium-sized receptive field loss. Medium-size receptive field loss may correspond to parvocellular (p-cell) processing deprecation. Our major conclusion provides a new measurement, using PSD analysis to assess mTBI conditions from primary V1 areas. The statistical analysis demonstrated significant differences between the mTBI and control cohort in the Visual Evoked Potentials (VEP) amplitude responses and PSD measurements. Additionally, the PSD measurements were able to assess the improvement in the mTBI primary visual areas over time through rehabilitation.

**Keywords:** keyword power spectral density analysis; visual evoked potentials; mild traumatic brain injury; magnocellular (MC) pathway; parvocellular (PC) pathway

## 1. Introduction

Given a neurological event such as a traumatic brain injury, damage to the peripheral visual system may affect the sensory-motor feedback loop in its ability to process peripheral vision process information, such as balance, movement, coordination and posture, for cognitive processing [1]. Additionally, the peripheral vision process combines this information with the kinesthetic, proprioceptive, vestibular and even tactile systems for the purpose of orienting and acting as a master organizer of these other processes [1]. Damage to the peripheral visual system, such as in a whiplash event, can cause significant dysfunction at the level of the midbrain, where visual-spatial awareness is transmitted through fibers from approximately 20% of the peripheral retinas of both eyes [1].

A study by Padula [2] found that a minimal 10 mile an hour rear-end collision is equal to 14,000 lbs. of inertial force when applied to the spinal cord and visual systems [2]. Although this type injury cannot be seen, in most cases, on a computerized tomography (CT) scan or Magnetic resonance imaging (MRI), Visual Evoked Potentials (VEP) have been found to capture instances from individuals who frequently experience a mild traumatic brain injury (mTBI)-related syndrome related to visual trauma [2]. According to the Centers for Disease Control (CDC), the incidence rates of mTBI range between a conservative 300,000 per year and a more liberal, recent estimate of 3.8 million cases in the United States annually [2,3].

Sports-related concussions represent between 15% and 20% of these injuries. American football is believed to account for a majority of these concussions [3]. Following an mTBI event, an individual will often experience a range of symptoms related to vision, such as diplopia, vertigo, asthenopia, inability to focus, reading/concentration issues and photophobia [4].

VEP responses enable an understanding of the effects on the architecture of the visual system during instances of mTBI. Transient evoked potentials are responses of the system to sudden changes (jumps or steps) in the input [5]. VEPs measure the processing of the visual system, which features two major pathways: the parvocellular (P) pathway, originating in the midget retinal ganglion cells (RGCs); and the magnocellular (M) pathway, originating in the parasol RGCs [5,6]. These pathways are associated with specific functions, conveying the "what" (visual areas (V1)-(V3)-(V4)-inferior temporal cortex (IT)) and "where" ((V1)- > (V2)- middle temporal area (MT)-superior temporal sulcus (STS)) of visual information [7,8]. Finally, the ventral stream is fed by the PC cells, whereas the dorsal stream is fed by the MC cells [9].

Magnocellular and parvocellular systems have differential sensitivity to object size when a checkerboard pattern modulates between low and high spatial frequencies [10–12]. The magnocellular system is particularly sensitive to large objects (low spatial frequency), whereas the parvocellular system is more sensitive to small objects (high spatial frequency) [13]. The study by Jindra and Zemon [13] demonstrates how contrast sensitivity, through the isolated squares from a checkerboard pattern, activates the magnocellular and parvocellular systems. Magnocellular neurons show greater sensitivity to low luminance-contrast stimuli, whereas high-contrast stimuli preferentially activate the parvocellular system [14]. The checkerboard stimulus presents a common strategy to preferentially activate the M and P pathways, where the M pathway responds to low in contrast, large in grid size, stimuli and reverses the contrast at a fast rate vs. the P pathway, which responds to high in contrast, small in grid size and reversals in contrast at a slow rate [14,15].

Electroencephalograph (EEG) recordings produce the evoked-related potentials (ERPs) of neural population behavior over the entire cortex, and visual evoked potentials (VEPs) demonstrate the neural activity within the occipital cortex area, as triggered by what the patient is viewing. ERP and VEP waveforms are often characterized by the magnitude of three peaks and troughs referred, to as "P1", "N1" and "P2". The P1 is a positive component that usually culminates between 80 and 120 milliseconds (ms) after stimulus, which originates from the middle occipital gyrus [16].

A longitudinal study by Freed and Hellerstein features significant VEP waveform abnormalities (P1 latency delay >15% and/or amplitude decrease >50% and/or difference >15% between left and right eyes) in cohorts who have received optometric treatment after mTBI (group I) and cohorts who have not received optometric treatment after mTBI (group II), as compared to cohorts without mTBI (Sensitivity: 78%, Specificity: 100%) [17]. Overall, 72% of the group I patients and 81% of the group II patients revealed VEP waveform abnormalities. Additional testing, performed 12–18 months later, showed that 38% of the group I mTBI patients continued to display VEP waveform abnormalities after receiving a treatment regimen of optometric rehabilitation, whereas 78% of the group II TBI patients demonstrated waveform abnormalities. This study showed that VEP waveform analysis in patients with mTBI can be a reliable tool for the objective assessment of visual system deficits and recovery.

Power spectral densities of EEGs have been used to address neurological deficits. Twenty-one male participants who were football athletes in high school, where the control group consisted of 14 volunteers (Age 15.86 ± 0.67 years) and the concussed group consisted of seven volunteers (Age 15.97 ± 0.74 years), were used in a study by Munia et al. [18]. The EEG data was captured per participant during trial sessions, which involved engaged tasks with eyes opened and eyes closed. The Power spectral density (PSD) was determined by computing the Fast Fourier Transformations (FFT) for 1 to 40 Hz frequency bins of each EEG channel. The frequency bins were separated in the following manner: delta (1–4 Hz),

theta (4–8 Hz), alpha (8–12 Hz), beta (12–30 Hz) and gamma (30–40 Hz). Significant differences were observed between the athletes who had a concussion and the non-concussed cohort during three conditions: vigilant task (VT), eyes open (EO) and eyes closed (EC), in the following frequencies: 1–3 Hz, 9–10 Hz, 27–30 Hz and 35–38 Hz. An increase in the theta band during the VT activities has been implicated as a condition that leads to depression and inattentiveness [19]. Conversely, a decrease in the beta waves may point to poor cognition and difficulty in concentration [20].

This study's aims and objectives are on the measurement of the responses of VEPs and their implications to the visual pathways from sports-related mTBI injuries, and introduces a new measurement featuring power spectral density (PSD), where the highest amount of power per frequency lies within a frequency spectrum (1–100 Hz) between the control and mTBI cohorts. We compare the VEP and the slope of the PSD measurements to demonstrate the robustness of our new methodology. Normal neural activity is stabilized by thresholds and refractory periods, which provide the local homeostasis that regulates the neural firing rates across the cortex [20–22]. We propose that imbalanced firing rates, where the dominant power is found in either lower or higher frequency ranges, can be quantified as pathological neural states, such as instances of mTBI, through the PSD measurements.

## 2. Materials and Methods

### 2.1. Cohort Description

Seven patients (1 male, 6 females; Mean Age = 25.56 years.) with different types of visual dysfunction due to mTBI were recruited in this pilot study (see Table 1). Seven healthy non-mTBI participants (1 male, 6 females; mean age = 23.71 years) were also recruited as baseline controls for this study. The research followed the tenets of the Declaration of Helsinki. Informed consent was obtained from the volunteers after an explanation of the nature and possible consequences of the study. This study acquired approval from the Institutional Review Board of the Southern College of Optometry (IRB #0000673, 21 August 2017). The diagnoses of the mTBI patients was established via a physician who specializes in sports-related concussion assessment and rehabilitation, and an optometrist specializing in visual evoked potentials and visual rehabilitation. The characteristics of the seven patients are summarized in Table 1 [23,24].

Mild traumatic brain injury patients were selected for this study by the head athletic trainers at one of three colleges in Memphis, including: The University of Memphis, Rhodes College and Christian Brothers University. The student athletes who had suffered a mild TBI were sent to The Eye Center at Southern College of Optometry (SCO) for an evaluation of their visual processing and performance as soon as they were medically stable, where VEP analysis is included in the workup protocol [24,25]. The visits all occurred within two weeks of their most recent concussion. Complete optometric clinical visual evaluations were conducted on all volunteers. The visual evaluations include: distance and near visual acuity (oculus dexter (OD), i.e., right eye; oculus sinister (OS), i.e., left eye; and oculus uterque (OU), i.e., both eyes; cover test at distance and near; motility testing; near point of convergence; reach grasp release testing; near and through challenge lenses; eye health testing; anterior and posterior segment testing; pupil testing; color vision testing; global stereo testing; near and distance retinoscopy; and a full analytical, including refraction. All of the volunteers had corrected visual acuity of 20/20 in the right and left eyes and all VEP testing was conducted with full correction in place. Volunteers would have been excluded had strabismus been found to be present.

The patients in this study were selected based on the mild severity of their brain injury, where more severe instances were excluded from this study, and visual lesions were detected via brain imaging, when necessary. Additionally, we were dealing mostly with diffuse types of injury, not focal lesions, such as brain tumors or other types of brain diseases. Vision is an end-to-end system. If there is a disruption anywhere in the primary visual system, it is likely to manifest throughout the visual circuitry. Mild traumatic brain injury, in this instance, was assessed through the primary visual responses from checkerboard

stimulus. Through this approach, we were looking to quantify the disruption to the primary visual pathways, which involve either medium-sized or large-sized receptive field loss.

**Table 1.** mTBI patients, injuries and symptoms.

| Volunteer Number/Age (Years)/Gender | Sport/Prior Head Injuries | General/Visual Symptoms |
|---|---|---|
| TBI 1/25–30 years/F | Cheerleader<br>10 prior concussions before current mTBI | • Headache<br>• Light headed<br>• Foggy mind<br>• Sleeping more and lethargic<br>• Unable to concentrate<br>• Photophobia<br>• Pain with eye movements |
| TBI 2/15–20 years/F | Soccer<br>2 successive concussions within 2 days.<br>Involved in car accident in 2015. | • Headache that never goes away<br>• Sound sensitivity<br>• Nausea without vomiting<br>• Loss of appetite<br>• Inability to sleep well or long<br>• Hard to focus on tasks<br>• Unable to study<br>• Vasovagal type postural hypotension leading to syncope |
| TBI 3/25–30 years/M | Rugby<br>>5 concussions | • Nausea<br>• Sensitivity to light—photophobia |
| TBI 4/20–25 years/F | Soccer<br>>5 concussions | • Headache<br>• Increased sleep<br>• Nausea<br>• Sensitivity to light—photophobia |
| TBI 5/40–45 years/F | Soccer<br>>5 concussions | • Headaches<br>• Blurry vision<br>• Light and noise bother her<br>• Cannot be around fluorescent lights.<br>• Word retrieval issues |
| TBI 6/15–20 years/F | Soccer<br>4 other confirmed concussions.<br>Car accident in 2017. | • Short term memory issues<br>• Lights and noise bother her<br>• If she runs, vision is blurry<br>• Exhibits short term memory loss |
| TBI 7/15–20 years/F | Basketball<br>>5 concussions | • Slept most of 1st week after concussion.<br>• Nausea after the 1st week<br>• Dizziness with school work<br>• Double vision<br>• Body feels heavy and groggy |

The visual evoked potentials were captured via a LKC Technologies Pattern UTAS VEP system to evaluate the positive peak of the waveform after 100 ms from stimulus, known as P1 [26,27]. The system is composed of two monitors; where one monitor presents a checkerboard stimulus pattern with a spatial frequency stimulus (8, 16, 32, 64, 128) to the patient and the other presents the VEP responses from the patient. The system has the capability to remove transient responses from the patient data, such as eye blinking and ambient machine noise (~60 Hz), via a notch filter.

The VEP recordings were obtained using gold cup electrodes (i.e., active, reference and ground) (Grass Technologies, Astro-Med, Inc., West Warwick, RI, USA), each 1 cm in diameter. The reference and ground electrodes were placed over the patient's ear lobes using ear clips, with the right ear lobe being used as the reference and the left ear lobe being used as the ground. The active electrode was placed over the primary visual cortical area Oz. The designated scalp regions were cleaned with alcohol wipes, and abrasive gel and

conductive paste were used to attach the electrodes. To maintain the electrodes firmly in place, an elastic head band was applied. Fifteen trials for each volunteer were performed. The test duration of each trial was 40 s, with the checks changing at a 1 Hz (2 reversals per second) temporal frequency. Conventional VEP test stimuli were employed. The screen size was 17 inches, measured diagonally, and was viewed from a distance of 1 m. Five different spatial frequencies were used, including $8 \times 8$, $16 \times 16$, $32 \times 32$, $64 \times 64$ and $128 \times 128$. These correspond to 112, 56, 28, 14 and 7 min of arc for our testing parameters. The mean luminance of the screen was 75 cd m$^{-2}$ and the checks were presented at 85% contrast. Electrode contact impedance was maintained <5 kΩ.

The rationale for measuring the monocular function (Oz) for both the right/left and binocular fields is based on the degree of differences between the channels OD vs. OS when patching between the eyes is applied. As an example, if the amplitudes of the VEPs exhibit a more than 10% difference between the OS and OD during 3 out of 5 spatial frequencies, then we may see, during binocular summation, a 10% or more binocular amplitude over the largest eye amplitude on 3 of 5 or more spatial frequencies over normal binocularity [28].

Dominant amplitudes from one eye will exhibit significantly reduced binocular amplitudes due to TBI. This may be due to interference from the dominant eye upon binocular visual integration, causing an overall reduction in the primary visual systems [28].

All of the testing was conducted in a room with the lights off. A lux meter held at the plane of the volunteer's face registered 0 lux with the computer screens turned off. Each of the 5 spatial frequencies were recorded with both eyes together, with the right eye alone and with the left eye alone [2,28]. An opaque patch was used when taking the monocular recordings. The order of all the trials for all the volunteers followed our standard clinical protocols and began with the binocular recordings, then the right eye recordings, ending with the left eye recordings. In all instances, we began with the $8 \times 8$ size and proceeded up each octave, finishing with the $128 \times 128$ recording. The volunteers were given 30 s breaks between the five recordings for each "eyed" condition and a full minute between eyes, while the patch was placed or moved. The total testing time was approximately 12 min in duration. A 0.25 ° radius red circle was presented in the center of the test field to control the fixation in order to maintain visual attention [28]. The volunteers were instructed to fixate upon this small central target with minimal blinking to reduce any response artifacts. The same protocol had been tested fully by the laboratory in the control groups and was applied to the mTBI cohorts [3].

### 2.2. Power Spectral Density Analysis

Brain activity analysis is performed through power spectral density (PSD) analysis of the EEG patterns, which provides the dominant power per unit of frequency within a range of frequencies, (i.e., 1–100 Hz) in units of power/frequency. The patient VEPs were exported from the LKC and processed via custom PSD routines coded in MATLAB® 2013 (The MathWorks Inc, Natick, MA, USA). A typical PSD display features a diminishing sinusoidal waveform pattern, where dominant frequencies reside in the beta-gamma brain frequency range (20–80 Hz). Linear regression is performed on the PSD values, where the slope is taken as it relates to the dominant power in decibels (dB) per frequencies within the cortex, in this instance in log10, as shown in Figure 1.

Directional shifts between high and low frequencies are associated with changes in the cerebral blood flow and metabolism [29,30]. The detection of state transitions in the EEGs is important in brain studies as it relates to the mechanisms of the loss of stability in neural populations due to brain disease or trauma [29,30]. When trauma occurs in the brain, stability among the neural populations is lost, and dominant low frequency patterns emerge and can therefore be detected [30].

The PSD in the normal state of human and simulated EEG conforms roughly to 1/f, with a slope between 2 and 3. High amplitude slow waves and a loss of the peaks in the PSD in the classical ranges (beta—gamma frequency ranges) would bring a shift in the PSD nearer to 1/f, with a slope near 4. We attribute these properties to a sustained loss of

the large-scale synaptic integration that normalizes by cooperative interactions through competitive inhibition and prevents the breakdown of the large-scale organization of the neocortex [11,31,32]. We would expect mTBI instances to cause dominant local neural activity, which would imply a diminished global integration through the sustained loss of long-range correlated activity, also seen in seizure activity due to the power in both the low and high theta ranges [31,32].

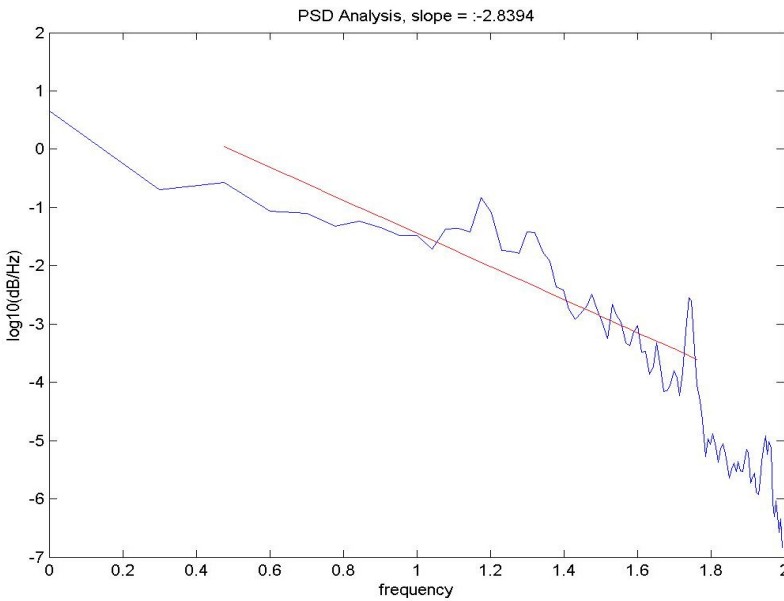

**Figure 1.** Linear regression analysis of PSD spectral slopes of the EEG spectrum of control participant.

### 2.3. Recording of VEPs

The checkerboard pattern stimulates the subpopulations of visual neurons tuned to spatial frequencies and orientations. Low spatial frequencies will appeal to neurons with large receptive fields. Neurons with medium-sized receptive fields will respond to higher spatial frequencies. These neurons need the least stimulation and produce larger VEPs. PSD analysis will show instances of either dominant low frequency or high frequency responses, or the loss of frequency responses due to the stimulus. As an example, if there is some degree of loss from large receptive fields from low spatial frequency stimulus, we would see a reported high PSD value (>3.5 dB/log10 Hz). The high value is due to a loss of power at the alpha frequency band (8–13 Hz), causing the slope across the frequency spectrum to shift the lower frequency upwards and the higher frequency downwards, thus causing a more negative slope.

### 2.4. Statistical Analysis

F-test two-sample for variances were used to analyze the monocular and binocular responses (OD, OS, OU) with the factors group (mTBI cohorts, control group) and checkerboard frequency stimulus (8, 16, 32, 64, 128). Although age was not correlated with the EEG measures, on mean, the mTBI patients' ages were ($25.56 \pm 8.5$ years) and the controls were ($25.76 \pm 1.5$ years). An a priori significance level of $\alpha = 0.05$ was used for all statistical testing. F-test two-sample for variances was used to analyze the VEP P1 data with the factors group (mTBI, control). Age was not considered as a covariate in the models for age-related changes in the visual evoked potentials. An a priori significance is calculated for all statistical testing, where 5 dependent variables times 3 viewing conditions (OD, OS and OU) were examined, which means multiple comparisons must be made, e.g., $\alpha = 0.05/15$, so $p < 0.003$ is used to reject the null hypothesis, so that there is a difference between the mTBI and control cohorts.

*2.5. Rehabilitation*

Following a mild TBI, the patients were provided active rehabilitation, such as visual therapy, which has been shown to improve faster outcomes to recovery. Visual therapy involves the following techniques: prism glasses, polaroid glasses and computerized vision training. Typically, visual therapy is a once-weekly 45–50 min in-office therapy, with home practice to be completed on the days between sessions. The sequence and the loading of the visual techniques are customized for each patient based on their needs. Visual therapy rehabilitation programs may finish in as little as six months, but most continue for longer periods of time, as long as there are meaningful unmet visual needs that continue to be served by more therapy.

## 3. Results

The following F-test results, shown in Table 2, were conducted on the VEP P1 values and revealed the following (F7,7). Measuring the mean and averaging the standard deviation (SD), the mean difference is 12−9 = 3 uV. The shared SD = 3.5, and Cohen's d = 0.3. Therefore, the power (two tailed test) = 0.315. Table 2 illustrates the significant differences between the TBI and control groups.

**Table 2.** VEP P1 *p*-values.

| | OD | OS | OU |
|---|---|---|---|
| 8 × 8 | 0.006 | 0.003 | 0.024 |
| 16 × 16 | 0.067 | 0.01 | 0.006 |
| 32 × 32 | 0.413 | 0.004 | 0.321 |
| 64 × 64 | 0.032 | 0.224 | 0.126 |
| 128 × 128 | 0.04 | 0.001 | 0.081 |

VEP P1 response (~100 ms) analysis was performed on seven controls and seven mTBI cohorts, with a mean and standard error between groups, as shown in Figure 2 and Table 3. The mTBI group exhibited a higher mean micro-volt range (10.5–17.5 µV), excluding the 128 × 128 frequency, which had a range of 6.5–8.5 µV. The control group exhibited a micro-volt range of 8.5–12.5 µV, with less variability.

**Table 3.** VEP mean and standard error values, mTBI values and controls.

| | mTBI | | | Controls | | |
|---|---|---|---|---|---|---|
| Frequency | OD | OS | OU | OD | OS | OU |
| 8 × 8 | 11.5 ± 1.9 | 13.5 ± 2.3 | 14.1 ± 2.4 | 9.70 ± 0.5 | 10.0 ± 0.5 | 9.8 ± 0.9 |
| 16 × 16 | 12.1 ± 2.5 | 14.3 ± 2.5 | 16.3 ± 4.0 | 12.27 ± 1.2 | 9.5 ± 0.7 | 11.1 ± 1.0 |
| 32 × 32 | 12.3 ± 2.2 | 12.1 ± 3.5 | 13.5 ± 1.8 | 8.92 ± 2.0 | 9.1 ± 0.8 | 11.2 ± 1.4 |
| 64 × 64 | 13.5 ± 2.6 | 14.2 ± 2.1 | 16.5 ± 3.1 | 9.75 ± 1.0 | 9.2 ± 1.4 | 11.0 ± 1.8 |
| 128 × 128 | 7.8 ± 0.5 | 9.7 ± 1.2 | 13.2 ± 2.4 | 10.02 ± 0.5 | 9.3 ± 0.8 | 11.5 ± 1.2 |

The VEP P1 mTBI mean *p*-values (7.8 ± 0.5–16.3 ± 4.0) are within a very wide range of values, as seen in Table 3. All of the VEP mean values for the controls are within the 9.1 ± 0.8–12.27 ± 1.2 range.

The following F-test in Table 4 was conducted on PSD values revealed the following (F7,7):

Table 4 PSD values demonstrate significant differences between the groups in 8 × 8 OD, 128 × 128 OS.

Analyses via power spectral density methodology from the control and from those individuals who have suffered through visual impairment from mild traumatic brain injuries have provided the following results, presented in Figure 3 and Table 4.

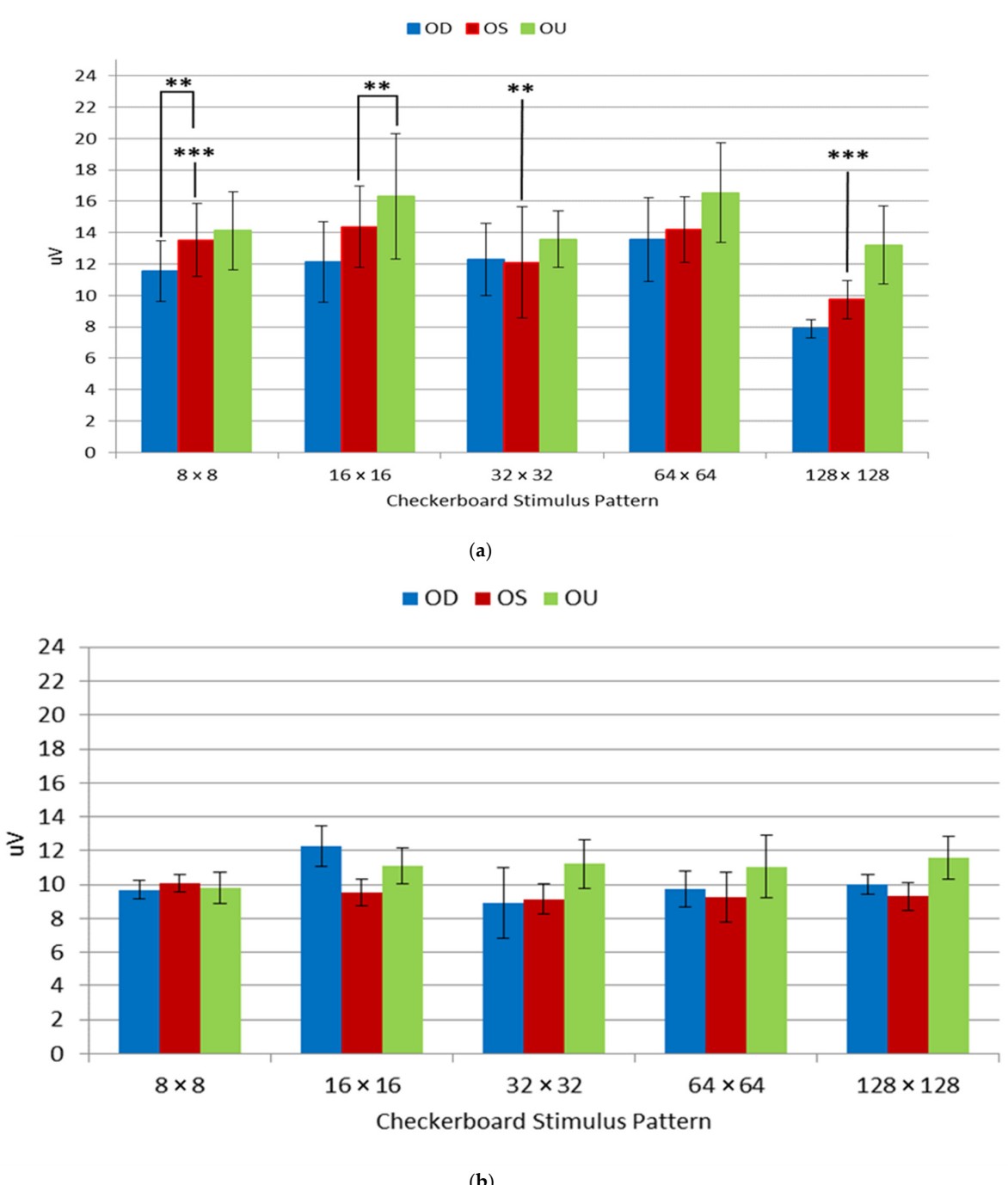

(a)

(b)

**Figure 2.** Display of mean visual evoked potentials (VEPs) values for oculus dexter (OD), i.e., right eye, oculus sinister (OS), i.e., left eye, and oculus uterque (OU) both eyes for mTBI (**a**) and controls (**b**). Significance thresholds designated at: ** = $p < 0.01$, *** = $p < 0.003$.

**Table 4.** PSD *p*-values.

| Frequency | OD | OS | OU |
|---|---|---|---|
| 8 × 8 | 0.001 | 0.089 | 0.398 |
| 16 × 16 | 0.04 | 0.009 | 0.029 |
| 32 × 32 | 0.186 | 0.016 | 0.136 |
| 64 × 64 | 0.169 | 0.01 | 0.079 |
| 128 × 128 | 0.012 | 0.003 | 0.103 |

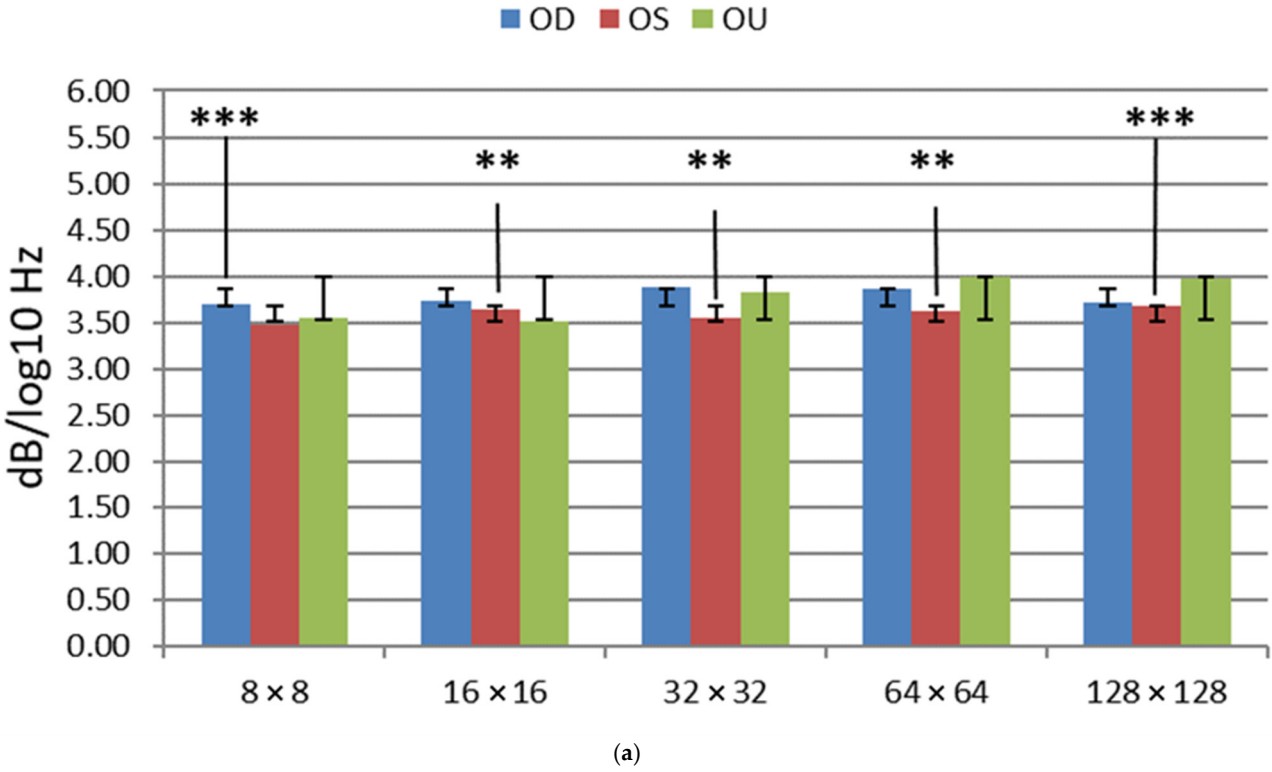

(**a**)

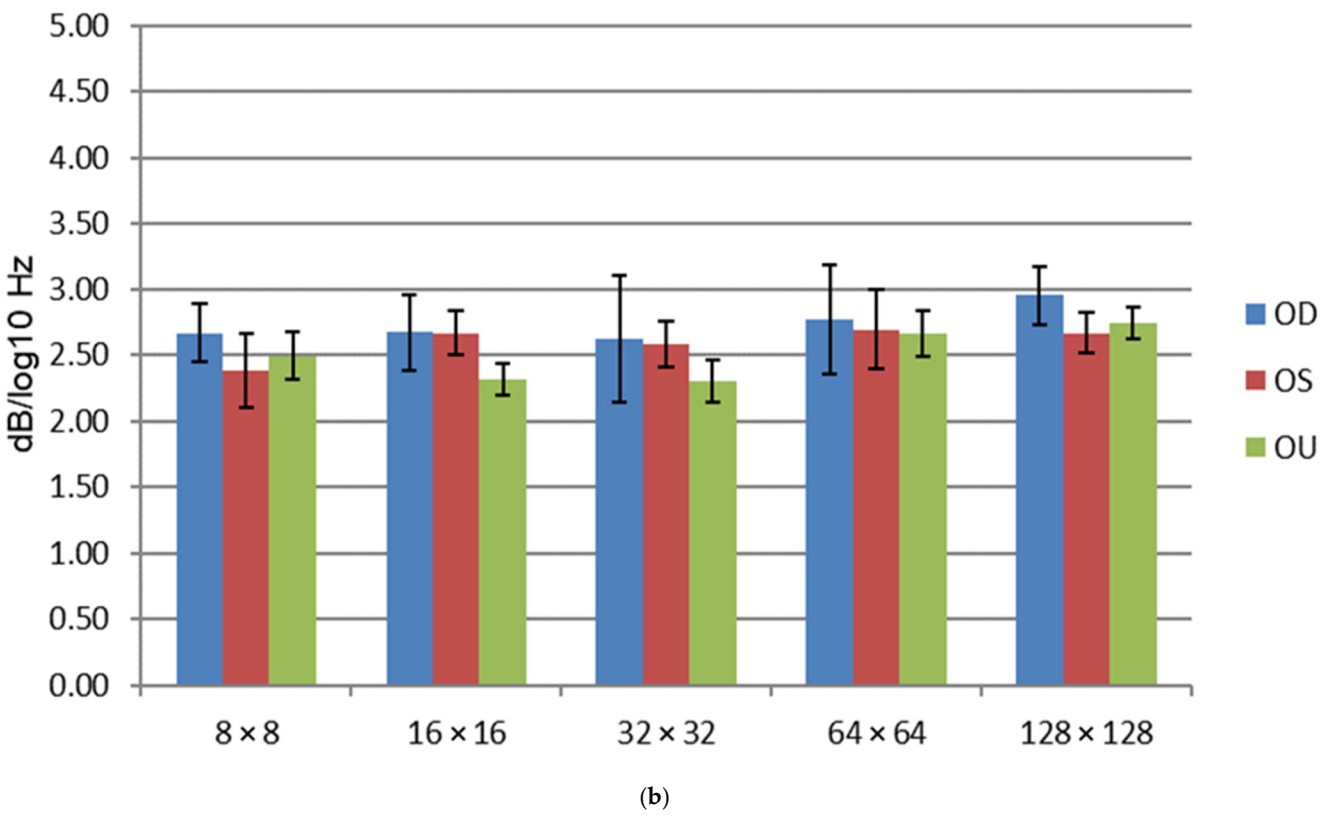

(**b**)

**Figure 3.** Display of mean power spectral density (PSD) values for OD, OS and OU for mTBI (**a**) and controls (**b**). Spatial frequencies are on the x-axis. Significance thresholds designated at: ** = $p < 0.01$, *** = $p < 0.003$.

The PSD mTBI mean *p*-values (3.54 ± 0.1–3.68 ± 0.2) are within a very tight range of values, as seen in Table 5. This is in stark contrast to the VEP P1 mTBI mean values. All of the PSD control mean values are within the 2.3 ± 0.1–2.9 ± 0.1 range.

**Table 5.** PSD mean and standard error values, (a) TBI values, (b) and controls.

| | mTBI | | | Controls | | |
|---|---|---|---|---|---|---|
| Frequency | OD | OS | OU | OD | OS | OU |
| 8 × 8 | 3.6 ± 0.2 | 3.48 ± 0.2 | 3.55 ± 0.2 | 2.6 ± 0.1 | 2.3 ± 0.1 | 2.5 ± 0.1 |
| 16 × 16 | 3.7 ± 0.1 | 3.65 ± 0.2 | 3.51 ± 0.2 | 2.6 ± 0.1 | 2.6 ± 0.1 | 2.3 ± 0.1 |
| 32 × 32 | 3.8 ± 0.1 | 3.54 ± 0.1 | 3.82 ± 0.0 | 2.6 ± 0.1 | 2.5 ± 0.1 | 2.3 ± 0.1 |
| 64 × 64 | 3.8 ± 0.2 | 3.63 ± 0.2 | 4.00 ± 0.3 | 2.7 ± 0.1 | 2.7 ± 0.1 | 2.6 ± 0.1 |
| 128 × 128 | 3.7 ± 0.2 | 3.68 ± 0.2 | 3.98 ± 0.2 | 2.9 ± 0.1 | 2.6 ± 0.1 | 2.7 ± 0.1 |

Through the rehabilitation techniques described in Section 2.4, Patient 2 and six showed a more significant improvement than the rest of the patients. There were no significant changes that could be quantified in patients 1, 3, 4, 5 and 7 through rehabilitation. Patient 2 presented very dominant large- and medium-sized receptive field loss across the OD, OS and OU, as seen in the high PSD values. The patient's medium-sized receptive field loss is slightly greater than his large receptive field loss, which may correspond to p-cell processing deprecation. The large reduction in the higher spatial activity demonstrates the restored p-cell processing. The PSD values for OD remain in the higher than normal range. The overall reduction over three months for Patient 2 is 6.37 dB/log10 Hz. The most significant reductions were found in the 64 × 64 OS and OU, and 128 × 128 OD, OS and OU, with a mean drop of 1.414 dB/log10 Hz ± 0.0668, see Figure 4.

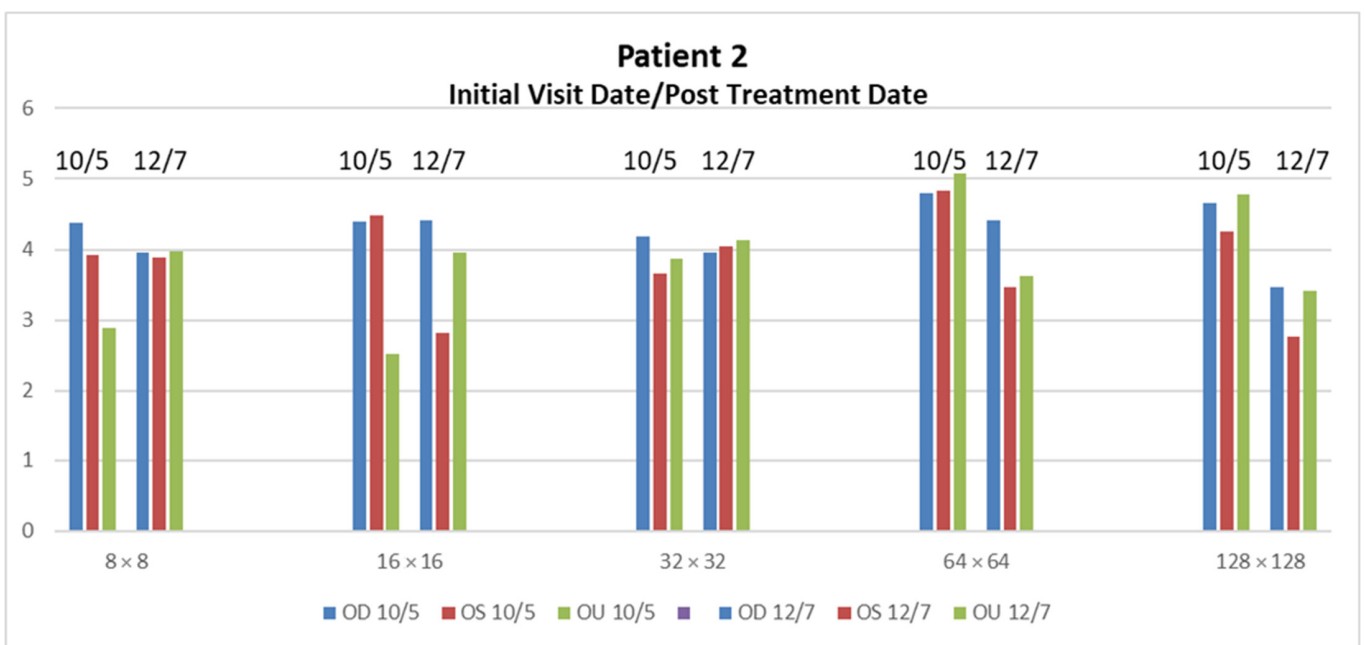

**Figure 4.** PSD reduction after two months in Patient 2 is seen in 64 × 64 OS and OU, and 128 × 128 OD, OS and OU. The restoration of PSD values for these specific areas within the 2–3 range have shown to contribute to the measurement of improvement of visual issues from mTBI.

Patient 6 demonstrated, overall, higher PSD pathological ranges due to mTBI visual issues specifically found in the 16 × 16 OU responses. Due to the relatively low trauma of this patient, visual treatment enabled this patient to be restored in a short amount of time. The overall reduction in three months is 7.438 dB/log10 Hz. The most significant reductions

were found in the 8 × 8 and 16 × 16 OU, with a mean drop of 0.775 dB/log10 Hz ± 0.0544, see Figure 5.

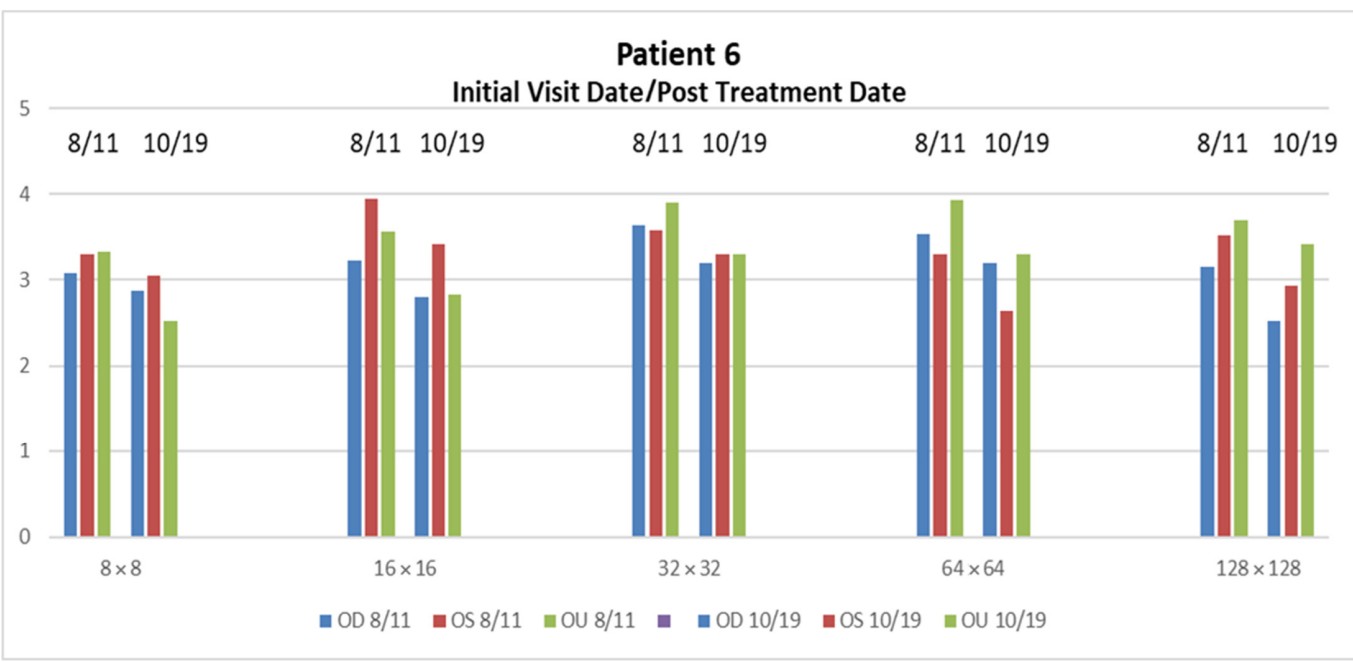

**Figure 5.** PSD reduction after two months in Patient 6 is seen 8 × 8 and 16 × 16 OU. The restoration of PSD values for these specific areas within the 2–3 range have shown to contribute to the measurement of improvement of visual issues from mTBI.

## 4. Discussion

Power Spectral Density slope values ($\alpha$) enable a correlation between the magnocellular and parvocellular activity in response to checkerboard spatial stimuli. In Figure 3a, the OU PSD slopes increase in relation to their matched OU PSDs by spatial frequency in relation to the OD and OS values. At a low spatial frequency, the OU is similar to the OD and OS responses. However, at the higher spatial frequencies, the OU PSD slope is the most negative of each of the groups. The results show as much as a 1.5 decibel/log10 (Hertz) separation between both populations. The PSD responses of mTBI reflect the same pathological values as seen from patients with other neurological issues, such as epilepsy, with an increased power in the delta-theta power, whereas the mTBI patients specifically exhibited a loss in power in the alpha range [11,31].

Patients exhibiting higher P1 VEPs with significance thresholds $p < 0.003$ are found in the 8 × 8 and 128 × 128 spatial frequencies in the OS. Additionally, the patients exhibited comparably statistically significant PSD differences between the control and cohorts are in 8 × 8 OD and 128 × 128 OS spatial frequencies. We can easily see the separation between the mTBI and control groups; as there is an almost 1.5 decibel/log10 (Hertz) separation between both populations. The most significant VEP instances, where $p = 0.003$, was in the 8 × 8 OS and 128 × 128 OS where $p = 0.001$. VEP instances where significance thresholds $p < 0.01$ included the following spatial frequencies, exhibiting the following values: 8 × 8 OD ($p = 0.006$), 16 × 16 OS ($p = 0.010$), 16 × 16 OU ($p = 0.006$) and 32 × 32 OS ($p = 0.004$). Figures 2 and 3, where significance thresholds $p < 0.01$, show that the following exhibit similar spatial frequencies: 8 × 8 OD, 16 × 16 and 32 × 32 OS, 16 × 16 OU and 128 × 128 OS. The PSD values conform to the VEP values, as seen in Figures 2a and 3a.

The analysis of the patients in the study found that there were additional instances of medium-sized receptive field loss, which may correspond to parvocellular (p-cell) processing deprecation versus magnocellular (m-cell) processing deprecation, based on the frequency stimulation. Patients 4 and 6 exhibited discernable loss across OU lower fre-

quency ranges, specifically around 8 × 8 and roughly 16 × 16 and 32 × 32, that probably contributed to the binocular issues. The patient's large-size receptive field loss may correspond to m-cell processing deprecation.

Patients 5 and 7's results demonstrated very dominant large- and medium-sized receptive field loss across the OD. There is also discernable loss across the OU that probably contributes to the binocular issues. The patients' medium-sized receptive field loss is slightly greater than the large receptive field loss, which may correspond to p-cell processing deprecation. In patients 3 and 1, we see a loss of medium-sized receptive field neurons due to the loss of theta-alpha activity that we would expect from higher spatial stimulus.

In the current study, we measured the cortical VEPs and spectral power in response to five spatial frequencies from the controls and from individuals who exhibited visual impairments from mTBI. The PSD values enable a quantification of mTBI instances compared to the analog waveform, which involves approximate min/max VEP measurements. The values seen in Figure 3a have a tighter range where the data resides compared to Figure 2a, which has a broader range due to the calculated values of the PSD waveform. Additionally, the standard error of the PSD is substantially minimized compared to the VEP waveform measurement due to the direct calculation of the power of the waveform over the log10 frequency.

Our work conforms to similar studies, where a rise in the slope from the lower frequencies portends instances of TBI and brain pathologies. In the study from Munia et al., twenty-six slope values from the power spectral density of the EEG data showed significant differences between athletes who had a concussion and the non-concussed cohort. Instances of increased power in the theta and alpha frequencies have been shown after a TBI occurrence [18]. In another study, where power spectral analysis has shown to be an effective approach to brain state quantification, power spectral analysis was performed on the visual evoked potentials to counter-phased checkerboard stimuli from 49 patients with multiple sclerosis (MS), where the consideration of both the PSA and latency of the VEP increased the percentage of MS patients exhibiting visual pathway conduction abnormalities. from 61% to 86% [33].

Several animal studies have utilized VEP analysis to quantify mTBI instances. A rat model was used to detect changes in the VEP waveform from two readings post-TBI incident [34]. In another instance, a mouse model of repetitive mTBI (rmTBI) was used to further characterize visual deficits using an optomotor task, electroretinogram and visually evoked potential, to locate likely areas of damage to the visual pathway [35]. Using stimuli similar to their previous study, the authors reported that a greater number of concussions were positively correlated with increased P300 latency [36].

Human studies featuring VEP analysis of mTBI instances have included decreased amplitudes and increased latencies of the P3b component after mTBI [37]. In concussed athletes that were symptomatic, significant reductions in visual P3 amplitude were reported compared to recently concussed, but asymptomatic, boxing and soccer athletes and never concussed athletes [38,39]. In addition, longer visual P3 peak latencies were correlated with self-reported attention and memory deficits in athletes with three or more concussions [40]. Additionally, there is some evidence that a history of three or more concussions is associated with changes in cognitive neurophysiology [40]. Furthermore, athletes with three or more concussions may be at an increased risk of sustaining a future concussion (86), and are more likely to have a slowed recovery [40].

The PSD analysis provided relatively similar results as the VEP analysis, where PSD analysis is derived by the slope of the linear regression of 1 to 100 Hz. VEP analysis is estimated at the maximum of the amplitude of the P1 waveform after the initial stimulation. Through the PSD analysis, we were able to observe a loss in power in the alpha range from the mTBI cohorts across all patients, which contributed to the high PSD values.

Due to the limitations of this study include the small data set and the varied sports-related injuries. Gathering cohorts who exhibited mild traumatic brain injuries that manifested into visual deficits was a challenge for this study. We broadened our cohort to

involve all sports activities, across several local universities and colleges. Even with this broad net, we only gathered less than two handfuls of cohorts, with a wide range of sports activities and age disparities and limited gender representation, yet the TBI effects were within a repeatable range of outcomes, i.e., headache, photophobia, nausea, memory issues and sleeping issues.

There have been instances during rehabilitation where the brain may undergo sensory reorganization by recruiting multisensory neurons after progressive degenerative visual conditions [32,37,38]. Additionally, it is conceivable that internal sensory reorganization may be an adjunct method for internal repair in instances of induced visual cortical injury. Neurosmithing is accomplished through active external rehabilitative techniques and the brain's set of internal repair mechanisms, which seemed to have occurred in patients 2 and 6. The striate and extrastriate visual areas may be utilized due to their involvement in visual imagery as internal visual stimuli for visual cortical reorganization. Additionally, internal cross-sensory stimuli can be marshalled to activate the primary visual areas, such as auditory stimuli, which challenge our idea that sensory stimuli may have more than one type of input [32].

We have found instances from mTBI that have caused a loss of alpha activity, which may implicate similar deficiencies in the attentional states found in the frontal-parietal networks. The alpha bandwidth has been associated with attentional states, specifically found in the frontal-parietal networks [7,13,38–42]. Alpha rhythms seem to be regulated during activities when an individual is focused on a given task [12]. The inability to modulate the alpha activity has been implicated in visual attentional deficit disorders [43–45].

The peripheral process provides spatial mapping to the cortex through postural orientation via the primary visual systems. A feedback mechanism to the thalamus, midbrain and brainstem occurs when the spatial mapping to the cortex changes based on new input upon postural orientation. Similar issues in the peripheral process of mTBI have been seen in those with lesions in the superior colliculus, affecting the match of peripheral visual information with sensorimotor information. A mismatch of peripheral visual information will affect fusion and binocularity, causing convergence insufficiency, divergence excess and exotropia [46,47]. Ketchum, et al. suggests that the deprecation in the proprioception systems will lead to a feed-forward mechanism that will affect the efficiency of movement [48].

## 5. Conclusions

Induced brain pathology in primary visual areas exhibits several unique characteristics. The VEP analysis has shown significant differences in the P1 responses in OD across most frequency stimuli. The PSD slope values ($\alpha$) provide quantifiable measurements vs. the VEP analysis that relates to the frequency-power distribution between the mTBI cohorts and controls. Through this measurement, we find that mTBI patients who exhibit visual issues have alpha band frequency loss in the primary visual areas. This study has also found more instances of medium-sized receptive field loss, which may correspond to parvocellular (p-cell) processing deprecation versus magnocellular (m-cell) processing deprecation, based on frequency stimulation. Utilizing this methodology, this approach could be applied to provide a quantifiable value that signifies the state of the primary visual areas for visual issues related to mTBI.

**Author Contributions:** P.H. examined and selected the patients for this study. M.H.M. performed the programming and data analysis. M.H.M., P.H. interpreted the results and wrote and approved the final manuscript. All authors have read and agreed to the published version of the manuscript.

**Funding:** This investigation was supported by an unrestricted grant from the College of Optometrists in Vision Development (COVD) Aurora, OH, to the Southern College of Optometry, Memphis, TN.

**Institutional Review Board Statement:** Each participant gave informed written consent in compliance with the Institutional Review Board of the Southern College of Optometry (IRB #0000673, 21 August 2017).

**Informed Consent Statement:** Informed consent was obtained from all subjects involved in the study.

**Data Availability Statement:** The data is available upon request.

**Conflicts of Interest:** The authors declare no conflict of interest.

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
