# Peer review of "A Pilot Investigation of Visual Pathways in Patients with Mild Traumatic Brain Injury"

_2035-8377, doi:10.3390/neurolint15010032_

Round 1

Reviewer 1 Report

Minor notes:

- please check once again whether all abbreviations are introduced correctly (eg .VEP);

- part of the literature references are not in brackets (lines 30, 32, 81, 223, 225, 439...)

- all literature references are not cited in the text

- The figure description should be on the same page as the figure.

- The figures (2,3) show which groups have significant differences, but this is not precisely drawn. Namely, it is necessary to center the marks above the groups being compared. In your case, somewhere the dashes extend to the edge of the column, and somewhere to the middle of the column - it is correct to connect the columns between which there are differences so that the edge of the bracket is above the standard deviation line (that is, the middle of the two columns).

- You have divided Table 3 into a) and b). Why didn't you rather add a row that says which group it is.

- In the explanation of figure 4, you do not say what the numbers above the columns (dates?) refer to.

Main objections:

Unlike the introduction, the discussion is not written in a logical sequence, clearly and comprehensibly so that those who do not directly deal with the described methods could follow it. It also contains the results of measurements before and after rehabilitation, which do not belong in the discussion but in the results.

You only have one measurement during rehabilitation, so it is not certain whether this result is really an improvement. Usually at least 2-3 follow-up measurements are needed to determine progress.

When explaining the results, some animal studies can also help you.

https://www.liebertpub.com/doi/abs/10.1089/neu.2021.0165?journalCode=neu

Author Response

Dear Reviewer,

Thank you very much for your review.  It will greatly improve the overall quality of the paper.  Here are the responses:

- please check once again whether all abbreviations are introduced correctly (eg .VEP);  updated acronyms references

- part of the literature references are not in brackets (lines 30, 32, 81, 223, 225, 439...)  I could not find these references.

- all literature references are not cited in the text Updated

- The figure description should be on the same page as the figure.  Updated

- The figures (2,3) show which groups have significant differences, but this is not precisely drawn. Namely, it is necessary to center the marks above the groups being compared. In your case, somewhere the dashes extend to the edge of the column, and somewhere to the middle of the column - it is correct to connect the columns between which there are differences so that the edge of the bracket is above the standard deviation line (that is, the middle of the two columns).

Reformatted display

- You have divided Table 3 into a) and b). Why didn't you rather add a row that says which group it is.  Added row

- In the explanation of figure 4, you do not say what the numbers above the columns (dates?) refer to.

Updated

Main objections:

Unlike the introduction, the discussion is not written in a logical sequence, clearly and comprehensibly so that those who do not directly deal with the described methods could follow it. It also contains the results of measurements before and after rehabilitation, which do not belong in the discussion but in the results.

Updated

You only have one measurement during rehabilitation, so it is not certain whether this result is really an improvement. Usually at least 2-3 follow-up measurements are needed to determine progress.

Subject participation was difficult in this study.  Additionally, the only two participants that had shown any significant improvement were highlighted in the results.

In a similar study to ours, the authors used a singular recording to differentiate between magnocellular (transient) and parvocellular (sustained) neural pathways (Poltavski et al., 2017)

Poltavski D, Lederer P, Cox LK. Visually evoked potential markers of concussion history in patients with convergence insufficiency. Optometry and Vision Science. 2017 Jul;94(7):742.

When explaining the results, some animal studies can also help you.

Several animal studies utilized VEP analysis to quantify mTBI instances. A rat model was used to detect changes in the VEP waveform from two readings post TBI incident [46]. In another instance, a mouse model of repetitive mTBI (rmTBI) was used to further characterize visual deficits using an optomotor task, electroretinogram, and visually evoked potential, to locate likely areas of damage to the visual pathway [47].  Using stimuli similar to their previous study, the authors reported that a greater number of concussions were positively correlated with increased P300 latency [48].

Reviewer 2 Report

The paper entitled "A pilot investigation of visual pathways in patients with mild Traumatic Brain Injury" is very interesting, the authors examined visual processing within primary visual areas (V1) in normal and visually impaired individuals who exhibit significant visual symptomology due to sport-related mild traumatic brain injury.

How was pre-trauma status assessed?

I recommend better explaining the characteristics of inclusion of the controls in the materials and methods, how was the absence of minor head trauma assessed?

Could be interesting to the evaluation of visually impaired subjects that play contact sports like boxing or martial arts. If it is difficult for the authors to improve the samples they can integrate the discussion by talking about the impact of the mechanism of production of minor head injury in various sports and its consequences.

Author Response

Dear Reviewer,

Thank you very much for your comments.  It will improve the overall quality of the paper.  Here are the responses:

How was pre-trauma status assessed?

Seven patients (1 male, 6 females; Mean Age = 25.56 years.) with different types of visual dysfunction due to mTBI were recruited in this pilot study (see Table 1).  Seven healthy non-mTBI participants (1 male, 6 females; mean age = 23.71 years) were also recruited as baseline controls for this study. 

The visual evaluations include: distance and near visual acuity (oculus dexter (OD), i.e. right eye, oculus sinister (OS), i.e. left eye, and oculus uterque (OU), i.e. both eyes), cover test at distance and near, motility testing, near point of convergence, reach grasp release testing, near and through challenge lenses, eye health testing, anterior and posterior segment testing, pupil testing, color vision testing, global stereo testing, near and distance retinoscopy, and full analytical including refraction.  All volunteers had corrected visual acuity of 20/20 in the right and left eyes and all VEP testing was done with full correction in place. Volunteers would have been excluded had strabismus been found to be present.

I recommend better explaining the characteristics of inclusion of the controls in the materials and methods, how was the absence of minor head trauma assessed?

See previous description

Could be interesting to the evaluation of visually impaired subjects that play contact sports like boxing or martial arts. If it is difficult for the authors to improve the samples they can integrate the discussion by talking about the impact of the mechanism of production of minor head injury in various sports and its consequences.

In concussed athletes that were symptomatic, significant reductions in visual P3 amplitude were reported compared to recently concussed asymptomatic boxer and soccer athletes and never concussed athletes [50, 51]. In addition, longer visual P3 peak latencies were correlated with self-reported attention and memory deficits in athletes with three or more concussions [52].  

Round 2

Reviewer 1 Report

The authors accepted the suggestions.
The work is improved and acceptable for publication.